# The Fully Mediating Role of Psychological Resilience between Self-Efficacy and Mental Health: Evidence from the Study of College Students during the COVID-19 Pandemic

**DOI:** 10.3390/healthcare11030420

**Published:** 2023-02-01

**Authors:** Lu-Lu Qin, Jin Peng, Man-Ling Shu, Xin-Yi Liao, Hong-Jie Gong, Bang-An Luo, Yi-Wei Chen

**Affiliations:** 1Department of Social Medicine and Health Management, School of Medicine, Hunan Normal University, Changsha 410013, China; 2Department of Mental Health, Brain Hospital of Hunan Province (The Second People’s Hospital of Hunan Province), Changsha 410007, China; 3Department of Neurology, Xiangya Third Hospital, Central South University, Changsha 410013, China

**Keywords:** general self-efficacy, mental health, psychological resilience, college student, COVID-19

## Abstract

Student populations are susceptible to the COVID-19 pandemic and may easy develop mental health problems related to their immaturity of psychological development and fluctuation of mood. However, little has been known about the effects of the pandemic on college students and the associated influencing factors. This study aimed to explore the role of psychological resilience as a mediator between general self-efficacy and mental health. A cross-sectional survey was conducted with 480 Chinese college students from 12 universities in Hunan province of China. The participants responded anonymously to the *Generalized Self-Efficacy Scale* (GSES), the Chinese version of the *Resilience Scale for College Students* (RSCS), and the 12-item *General Health Questionnaire* (GHQ-12). Hierarchical linear regression and structural equation modeling were used in this study. The average of GSES and RSCS scores of college students were 25.00 ± 4.68 and 137.97 ± 15.50, which were at a medium level. The average score for the GHQ-12 was 1.59 ± 1.59, and 22.03% of the college students scored ≥ 3 on the GHQ-12, indicating that they were at risk of developing mental disorders. According to the analyses of mediation effect, psychological resilience played a fully mediating role in the relationship between general self-efficacy and mental health. In conclusion, Chinese college students were at high risk of developing mental disorders during the COVID-19 period. General self-efficacy was positively associated with psychological resilience, and psychological resilience played a fully mediating role in the relationship between general self-efficacy and mental health. Future studies and interventions should aim to promote psychological resilience and general self-efficacy.

## 1. Introduction

A worldwide pandemic of a novel coronavirus infection (COVID-19) triggered a severe epidemic of acute infectious pneumonia [1]. The World Health Organization (WHO) proclaimed that the COVID-19 pandemic is a very severe public health emergency, and the situation has received high levels of international attention [2]. College students, one of the populations who are most prone to anxiety, depression, and other mental health disorders, had an unprecedented risk of developing psychological illness during the period of the COVID-19 epidemic [3]. Studies have explored the psychological problems of undergraduates against the background of the COVID-19 epidemic [4,5], indicating that about 24.9% of students in colleges and universities in China have experienced anxiety symptoms on account of the COVID-19 epidemic [6]. Moreover, the mental health crisis caused by the infection disease continues and remains far from over [7,8].

The period of COVID-19 requires health departments not only to heal those infected sick, but also to treat the mental health problems caused by the pandemic. In addition, the pandemic emphasized in particular the significance of learning about management and control measures from countries and areas which have regular experience in responding to epidemics of infection diseases and other emergencies. However, studies related to the pandemic of COVID-19 globally have primarily paid attention to the epidemiology of the disease and the characteristics of patients, with less focus on the effects of the pandemic on mental health in undergraduates. Therefore, the generation of evidence that advances the objectives of global mental health within the context of the pandemic is vital, and it is particularly significant to explore the factors that influence and predict the mental health of undergraduates. Although the COVID-19 pandemic has led to an increased risk of mental health disorders [3,4,5,6], research on the relationship and its potential mediating factors has been limited.

Self-efficacy, the belief that an individual is able to accomplish difficult work or a challenging task and achieve one’s goals, reveals how confident people are in their ability to use their own skills to deal with life’s stress [9]. Existing studies have illustrated that there is a significant correlation between self-efficacy and mental health [10,11,12,13]. For example, people who perform better in self-efficacy are more likely to take healthy actions [13]. Therefore, the higher the level of positive factors, the lower is the associated level of negative factors [9]. Research also shows that self-efficacy is strongly associated with psychological health, and people who perform higher self-efficacy have better psychological health status [10,11]. A positive relation between self-efficacy and psychological health status has been indicated by correlation analysis [12]. The sense of self-efficacy in college students is able to reflect their levels of psychological health. People with strong self-efficacy are likely to be in a condition of good health [13].

Psychological resilience (PR) reflects the capacity of individuals to adapt to difficult situations, recover from painful injuries, and avoid possible harmful impacts after suffering from great adversity [14]. Psychological resilience, which improves people’s adaption, is a kind of positive personality [14]. Ong et al. illustrated that positive coping and adaptation when people confront injury, disaster, or difficulty is a connotation of resilience [15]. Lazarus and Folkman built an appropriate model to study transactional theory of stress [16], which is suitable for understanding health outcomes in college students during this epidemic. The framework indicates that it is critical to research the psychological resources of individuals to maintain mental health, especially in an adverse environment [16].

Self-efficacy and psychological resilience have been considered two of the key psychological resources when individuals are exposed to adverse conditions [17]. Studies have found significant positive relationships between self-efficacy and resilience, such as in freshmen nursing [18] and pregnant women [19]. The findings from Tan’s study indicated that resilience had a strong and powerful effect on mental health in the period of the COVID-19 pandemic [20]. The result is consistent with previous research [21,22] revealing that psychological resilience is an important predictor of mental health specifically in students in colleges and universities [21]. Furthermore, a study showed a positive association between resilience and mental health in nursing students in a college in China [22]. However, little is known about the effect mechanism of the three variables (psychological resilience, self-efficacy, and mental health).

Mental health is a comprehensive concept, involving stress, depression, anxiety, general self-efficacy, psychological resilience, post-traumatic growth, deliberate rumination, and so on. Evidently, the complexity inherent in identifying mental problems has been recognized as an important healthcare issue, and mental health problems may lead to substantial personal suffering [23,24,25]. Considering the threat to public mental health from COVID-19, many studies are working on improving mental health in this context. Despite studies having researched certain aspects of mental health, such as creativity, self-efficacy, and psychological resilience [25,26,27,28], no study has yet reported the relationship between psychological resilience, self-efficacy, and comprehensive mental health.

Compared with focusing on certain aspects of mental health, it might be more significant to understand the impact of COVID-19 on the comprehensive mental health status of individuals, which could provide reference for future comprehensive mental health intervention measures. In order to understand the effect of COVID-19 on comprehensive mental health, we used the *General Health Questionnaire* (GHQ) to assess students’ mental health. The GHQ is a self-administered screening tool designed to detect current mental disturbances and disorders [25,29]. Since its development by Goldberg and Hillier [29], the GHQ has been translated into 38 languages, a testament to the validity and reliability of the questionnaire [30], and research has reported that the GHQ was deemed the best validated screening tool for assessing mental health [31,32,33].

Due to the deficiency of study on the effect paths among the three variables (psychological resilience, self-efficacy, and mental health), and its importance, it is necessary to reveal how the above factors affect each other. Thus, based on previous studies, this research hypothesized: Firstly, that self-efficacy would positively predict the mental health of undergraduate population during the period of the COVID-19 pandemic; secondly, that self-efficacy is relevant to psychological resilience; and finally, that the self-efficacy of undergraduate population during the period of the COVID-19 is likely to affect their psychological health through the mediating effect of psychological resilience.

Student populations are susceptible to the COVID-19 pandemic and may easy develop mental health problems related to their immaturity of psychological development and fluctuations of mood. Therefore, this study aims to explore the effect mechanism of general self-efficacy, psychological resilience, and mental health in the population of Chinese college students during the period of the COVID-19 epidemic. The findings are useful for promoting recognition of the adverse impacts on mental health caused by the COVID-19 pandemic on the undergraduate population, and have more implications for psychological counseling services in the future.

## 2. Materials and Methods

### 2.1. Sample Size Calculation, Study Design and Procedures

This study involved a cross-sectional survey conducted in the Hunan province of China. A cross-sectional study formula was used to calculate the sample size: α = 0.05, n = u_α/2_^2^P(1 − P)/d^2^, where u = 1.96 when α = 0.05, and P represents the average incidence rate of anxiety symptoms, depressive symptoms, and somatic symptoms, which was 31.53% during the period of COVID-19 [34,35,36,37]. Considering the effective rate and response rate of the questionnaire, we needed a sample size of at least 445 participants.

By using multi-stage stratified cluster sampling, a representative sample of college students in the Hunan province of China were selected. Firstly, one third of the 36 universities in Hunan province were randomly selected, and the students were divided into five levels according to their grade. Then, two academic majors were randomly selected from each level of each school, and one class was selected randomly from each academic major. Our trained investigators invited students who were willing to participant in this study, after carefully introducing our research purpose and content. Data were collected through self-completed questionnaires, which were distributed and collected by researchers in classes on site. With the approval of the Research Review Committee of the Brain Hospital of Hunan Province (NO.2020126), our study was conducted from December 2020 to January 2021, which could be considered within the COVID-19 period. Finally, a total of 480 participants from 11 academic majors in 12 universities were recruited.

### 2.2. Socio-Demographic Information

The socio-demographic information collected in this study included age, gender, grade, home residence (urban vs. rural), economic status (below medium vs. medium vs. above medium).

### 2.3. Measures

#### 2.3.1. General Self-Efficacy

The revised Chinese version of the Generalized Self-Efficacy Scale (GSES) consisting of 10 items was applied to assess the participants’ levels of general self-efficacy, which included a four-point scale (1 = not at all true; 2 = somewhat true; 3 = mostly true; 4 = exactly true) [38]. Previous studies among Chinese undergraduate students have confirmed that the GSES demonstrates adequate reliability, and its obtained Cronbach’s alpha coefficient was 0.88 [38,39,40].

#### 2.3.2. Psychological Resilience

The Chinese version of the Resilience Scale for College Students (RSCS), constructed by Liu in 2007 [41], was applied to measure psychological resilience directly, and is suitable for use among Chinese university students. The RSCS consists of 38 items, ranging from 1 (very unlike me) to 5 (very like me). Higher scores indicate better psychological resilience. The reliability coefficient of the RSCS was 0.91. The test–retest reliability coefficients were 0.90, 0.92, 0.90, 0.86, 0.84, 0.75, 0.82, and 0.73 for the eight factors of Social Interaction, Family Support, Friend Support, Optimistic Tolerance, Self-worth, Self-recognition, Self-control, and Self-admission, respectively, and its Cronbach’s alpha was 0.92 [41].

#### 2.3.3. Mental Health

We assessed mental health using the 12-item General Health Questionnaire (GHQ-12). The Chinese version of the GHQ-12, revised by scholar Yang [42], is a widely used self-rating survey tool for psychological epidemiological investigation and the detection of mental disorders. Three factor categories were extracted from the GHQ-12, namely somatic symptoms, anxiety and worry, and depression or poor family relationships. According to the scoring method recommended by WHO, scoring for each question was counted as 0 or 1 point. The lower the total score within the range of 0 to 12 points, the better is the respondent’s mental health, and the higher the score, the higher their possibility of mental disorders [43]. Evidence shows that the GHQ-12 has good reliability above 0.79 in the Chinese population [44,45]. In the present study, the Cronbach’s alpha was 0.79.

### 2.4. Statistical Analysis

We conducted descriptive analyses of the sociodemographic variables, the t test, variance analysis, and Pearson correlation analysis of the key variables using SPSS 20.0 (IBM Corporation). Before the analysis, we tested all data for normality and found that it met the criteria. General demographic information that had an impact on relevant variables was used as the control variables for hierarchical linear analysis, and the Enter method was adopted. With mental health as the dependent variable and general self-efficacy as the independent variable, Model 1 was established. With psychological resilience as the dependent variable and general self-efficacy as the independent variable, Model 2 was established. Then the Model 3 had mental health as the dependent variable, general self-efficacy and psychological resilience as the independent variable. The common method of bias testing was used for analyses in the present study. Hierarchical linear regression was employed to examine the mediating role of psychological resilience in the association between general self-efficacy and mental health. Exploratory factor analysis including all items in the GSES, RSCS, and GHQ-12 was conducted to test for common method variance on the basis of the Harman’s single factor test. 

We used the Bootstrap method in the AMOS 22.0 (IBM Corporation, New York, USA) structural equation model to further verify the mediating effect [46]. We selected the following fitting indices to evaluate whether the model fitted the data: absolute adduct index including root means square error of approximation (RMSEA) and goodness-of-fit index (GFI), and value-added adduct including comparative fit index (CFI) and normal of fit index (NFI). We considered that GFI, CFI, and NFI values greater than 0.90 and RMSEA values are less than 0.08 [47] were acceptable. A 5000 bootstrap sample was utilized to calculate the indirect effects and their 95% confidence intervals (CI). If the 95% CI excluded 0, we recognized that the mediating effect would be significant [48].

## 3. Results

### 3.1. Characteristics of the Study Population

A total of 454 college students provided complete responses to our survey, contributing to an effective response rate of 94.6% (454/480). The college students’ age ranged from 17 to 24 years (20.20 ± 1.27). We recruited 154 males and 300 females. Among the 454 college students, 193 were from rural areas, while 261 were from urban areas in the Hunan province of China. About 62.8% (285/454) of the participants reported their economic status as medium.

### 3.2. The GSES, RSCS and GHQ-12 of the College Students

The average GSES score of the college students was (25.00 ± 4.68), which is close to the median value (26.00), indicating that their sense of self-efficacy was at a medium level.

The average score for the RSCS was (137.97 ± 15.50), which is close to the median (140.00), indicating that the participants’ psychological resilience was at a medium level.

The average GHQ-12 score of the participants was (1.59 ± 1.59). Based on the calculation for positive detection of mental disorder (total score ≥ 3 points), the scores obtained from the GHQ-12 showed that 22.03% of the college students were at risk of mental disorder.

The comparative results of each scale score among college students with different demographic characteristics are shown in Table 1. The GSES scores of female students were lower than male students (*p* < 0.05). The RSCS scores of male students were lower than female students *(p* < 0.05), the RSCS scores of students with low economic status were lower than students with medium economic status or above (*p* < 0.017, the *p* was corrected according to Bonferroni analysis). The GHQ-12 score at the age of 22 was lower than for students of other ages (*p* < 0.005, the *p* was corrected according to Bonferroni analysis), and rural students’ GHQ-12 score was higher than urban students (*p* < 0.05) (Table 1).

### 3.3. Correlation Coefficients of Self-Efficacy, Psychological Resilience, and Mental Health among College Students

Table 2, Table 3 and Table 4 show the results of the correlation coefficients of the variables. Correlativity among most variables was statistically significant. The exceptions were “depression or poor family relationships” with “social interaction” (*p* > 0.05) and “self-admission” (*p* > 0.05). The dimensions of mental health scores were obviously and negatively associated with general self-efficacy scores (*p* < 0.001) and psychological resilience scores (*p* < 0.001), while the psychological resilience scores were notably and positively connected with general self-efficacy scores (*p* < 0.001).

### 3.4. Hierarchical Linear Regression Analysis of Mental Health

The linear regression models of mental health are shown in Table 5. Model 1 revealed that the regression coefficient of general self-efficacy on mental health was statistically significant. Model 2 showed that the regression coefficient of general self-efficacy on psychological resilience was statistically significant, and Model 3 showed that the adding of psychological resilience observably cleared the impact of general self-efficacy on mental health.

### 3.5. The Mediating Role of Resilience in the Relationship between General Self-Efficacy and Mental Health

The mediating role of psychological resilience in the connection between general self-efficacy and mental health is represented, and the normalized path coefficients are shown by one-way arrows in the structural equation model (Figure 1). When psychological resilience was regarded as the intermediary variable, the path coefficient between general self-efficacy and mental health was markedly reduced, which confirmed the full mediating role of psychological resilience in the relationship between general self-efficacy and mental health. While the initial consequence was unsatisfactory, a revised model generated an approved model fit [49] (CMIN/DF < 5; RMSEA = 0.070; GFI = 0.943; CFI = 0.934; NFI = 0.908). According to the bias-corrected and accelerated bootstrap test, psychological resilience had an obvious mediating effect on the association between self-efficacy and mental health.

Table 6 indicates that the direct model (linking general self-efficacy to all dimensions of mental health) showed a non-statistically significant effect (β = −0.002, 95% CI: −0.015 to 0.011, *p* > 0.05). Moreover, the full mediation model (linking general efficacy to mental health through all dimensions of psychological resilience) had a significant effect (β= −0.018, 95% CI: −0.029 to -0.010, *p* < 0.000). Therefore, the total impact of general efficacy on mental health was significant (β= −0.020, 95% CI: −0.031 to -0.011, *p* < 0.000). Conclusively, psychological resilience played a full mediating role in the relationship between general self-efficacy and mental health.

## 4. Discussion

This study provides evidence that psychological resilience plays fully mediating roles in the relationship between general self-efficacy and mental health. These findings could enrich our understanding of the mechanisms of mental health, and may have implications for university mental health services during the COVID-19 pandemic.

As an acute, large-scale outbreak of infective disease, the COVID-19 pandemic has caused not only damage to individuals’ physical health, but also brought a significant impact on their mental health [4,5,6,7,8]. According to Bell’s research [43], a GHQ-12 score ≥3 is considered to represent risk of mental disorder. In our study, 22.03% of the college students scored ≥3 on the GHQ-12, indicating that they were at risk of developing mental disorders, which is similar to the results of previous studies [50,51]. Ma et al. indicated that approximately 45% of participants had psychological health difficulties, and the frequency of acute stress was 34.9% during the COVID-19 pandemic [40], while Wang et al. reported that prevalence of suicidal intention among college students was 9.2% during the non-epidemic period [51]. During the COVID-19 pandemic, students’ mental health was influenced by stressful issues related to the pandemic, such as the threat of COVID-19 disease, the disruption of daily routine, long-term social distancing, the challenges of attending online courses, and so on. Long-term social distancing during the epidemic may have changed the psychological state of college students, making them more likely to be anxious and depressed.

Furthermore, changes in interpersonal communication due to the pandemic would have an adverse effect on students’ mental health, and the pressure on college students in terms of study or employment was increased by the interference of the pandemic. Some college students experienced anxiety because of the low efficiency of online courses and their failure to maintain progress after the resumption of classes, while the severe employment situation in the post-epidemic era also aggravates graduates’ psychological burden. Moreover, Xiao et al. suggested that the pandemic may play a lasting negative role in college students’ mental health [52]. Therefore, policy makers and educators should provide timely and effective interventions for the mental health of college students during the period of COVID-19.

Our study indicated that college students’ general level of self-efficacy positively predicted their mental health status during the COVID-19 period, and that general self-efficacy also positively predicted their psychological resilience. More importantly, levels of psychological resilience were found to mediate fully the effects of general self-efficacy on their mental health. The first hypothesis was supported by our results confirming that the level of college students’ general self-efficacy positively predicted their mental health levels. This finding was based on Cervone’s cognitive model [53].

A study focusing on Iranian university students reported that a higher sense of self-efficacy can prevent stress and promote individuals’ psychological well-being [54]. This is in accord with studies by Lin and Wen [55,56]. Higher levels of general self-efficacy have been discovered to correlate with lower anxiety [57], reductions in depressive disorder [58], and greater optimism among undergraduate students [59]. Research reveals that general self-efficacy can affect individuals’ mood [60], and college students who had higher levels of general self-efficacy might demonstrate stronger self-confidence in various circumstances, which could help to lower the degree of worry and anxiety experienced during the COVID-19 period. Compared with higher general self-efficacy, lower self-efficacy might reduce an individual’s adaptability to the environment, and make them more prone to worse psychological well-being in difficult situations.

With respect to the second hypothesis, our study found that higher levels of general self-efficacy predicted higher levels of psychological resilience. This is in line with prior studies [61,62]. Interestingly, research found that resilience in entrepreneurial college students can be improved by self-efficacy [63]. Through the activation of affective, motivational, and behavioral mechanisms in difficult situations [64], entrepreneurial college students can improve their self-efficacy through continuously obtaining new knowledge, and then face more rationally the health threat brought by the epidemic, which is beneficial to the improvement of psychological resilience during COVID-19 control periods [63]. The current study indicated that psychological resilience had a protective influence on mental health, which is consistent with previous research [65]. Preceding studies also found that psychological resilience might have a critical effect on facilitating psychological health in various populations during the COVID-19 pandemic, such as medical and university students [66,67,68]. A study from Hong Kong focusing on undergraduate students revealed that more resilient individuals had lower rates of depression [69]. Recent studies found negative associations between resilience and worse psychological well-being (anxiety, depression) [70,71].

When the variables of psychological resilience were added to the model, the coefficient of general self-efficacy to mental health decreased, indicating that psychological resilience played a completely mediating role in the connection between general self-efficacy and mental health. We found support for the final hypothesis [72], documenting that resilience was positively correlated with self-esteem and life satisfaction, and negatively correlated with depression. Samuelson et al. [25] verified that greater general self-efficacy was connected with better mental health, suggesting that communities should target general self-efficacy to enhance psychological resilience during a pandemic. Arima et al. [73] indicated that schools should enhance self-efficacy to provide support for mental health, with a focus on improving resilience. The current results are also supported by those published by Ma et al. [19], which verified that resilience is a vital mediator of self-efficacy and anxiety among pregnant women. Hence, psychological resilience could be a good mediator between general self-efficacy and the mental health of college students in the COVID-19 period, and general self-efficacy can play an important role in promoting mental health through psychological resilience. Our study revealed that training to enhance students’ general self-efficacy and psychological resilience would be beneficial for university students by enabling them to confront the COVID-19 period more positively, which might their levels of depression to decrease and allow them to maintain good levels of mental health.

## 5. Limitations and Contributions

The current findings are tempered by the exploratory nature of the analyses and the small sample size. Firstly, all data were taken from Hunan province, China, from only 480 samples, which limits the extension of the research results. Secondly, we used a cross-sectional design to conduct the investigation, which limited our capacity to make causal inferences, and we cannot exclude the possibility that the self-reported survey design introduced bias. In future, further studies are therefore required to confirm these findings. These future studies can use longitudinal methods and/or randomized control designs to verify the causal relationships. Meanwhile, attention should be paid to changes in psychological resilience, self-efficacy, and mental health during the pandemic. Based on these results, more findings from studies focusing on the changes and causal relationships relating to this topic during the pandemic will be beneficial for preventing mental health problems for students in the future.

However, to our knowledge, this has been the first study to evaluate the relationship between general self-efficacy and mental health through the mediating effect of psychological resilience in college students during the period of COVID-19, and the results are able to provide a new perspective on how general self-efficacy affects mental health. These findings have bridged the research gap and provided new horizons for research in the area of mental health, and provide a basis for improving the mental health of individuals through the mediating effect of psychological resilience.

In terms of practical value, the results of this research have a vital role to perform in maintaining the psychological health of undergraduates during the period of epidemic. Thus, in view of the results from this study, we put forward several suggestions for universities and parents. Firstly, more attention should be paid to college students’ mental health, psychological resilience, and self-efficacy, especially during the pandemic and post-epidemic era. Secondly, considering the effects of the three variables, intervention strategies should be work together to provide a better scientific basis for the healthy growth of college students.

## 6. Conclusions

The Chinese undergraduate population was at high risk of developing mental disorders in the period of the COVID-19 pandemic, as 22.03% of them reported a score ≥ 3 on the GHQ-12. General self-efficacy was positively correlated to psychological resilience, and psychological resilience played a fully mediating role in the relationship between general self-efficacy and mental health. Providing training to enhance undergraduates’ general self-efficacy is likely to advance the necessary skills and beliefs to promote resilience, which can play an effective role in maintaining and improving the mental health status of college students at this time. The results of this research are of great significance for mental health education during the COVID-19 period. Future studies and interventions should be designed to promote psychological resilience and general self-efficacy.

## Figures and Tables

**Figure 1 healthcare-11-00420-f001:**
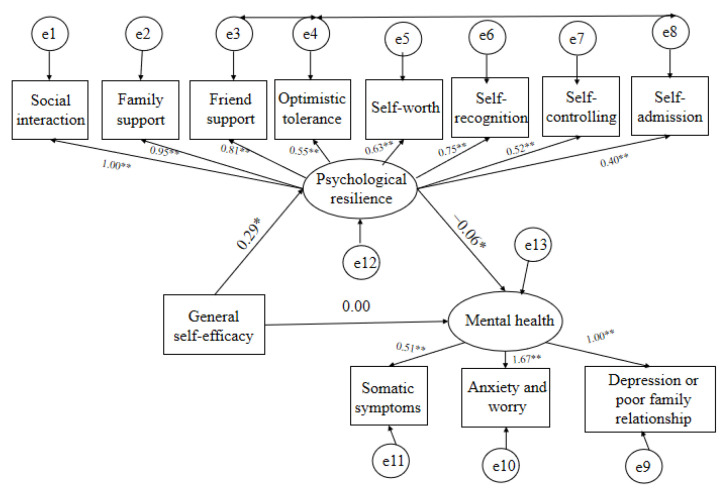
The mediating role of psychological resilience in the connection between general self-efficacy and mental health. (* Normalized path coefficients are shown on one-way arrow paths; the coefficient of the path is significant at the *p* < 0.05 level. ** All loadings in the measurement model were significant at the *p* < 0.05 level.)

**Table 1 healthcare-11-00420-t001:** Demographic characteristics and the distributions of variables among college students (*n* = 454).

Variables		Cases (N, %)	GSES Score (Mean, SD)	RSCS Score (Mean, SD)	GHQ-12 Score (Mean, SD)
Gender					
	Male	154(33.9)	25.97(4.39) ^a^	135.37(16.37) ^a^	1.69(1.86)
	Female	300(66.1)	24.50(4.75)	139.30(14.89)	1.54(1.43)
Age (years)					
	≤18	51(11.2)	24.96(5.17)	139.24(15.25)	1.35(1.64)
	19	52(11.5)	25.73(4.04)	139.33(14.50)	1.27(1.21)
	20	158(34.8)	24.46(4.62)	135.39(13.51)	1.78(1.58)
	21	145(31.9)	25.14(4.95)	138.75(15.69)	1.79(1.76)
	≥22	48(10.6)	25.63(4.05)	141.27(15.60)	0.96(1.11) ^b^
Grade					
	First year	62(13.7)	25.56(4.96)	139.55(15.53)	1.37(1.55)
	Second year	29(6.4)	24.90(4.34)	132.69(16.59)	2.21(1.88)
	Third year	199(43.8)	24.49(4.64)	137.48(14.98)	1.59(1.45)
	Fourth year	154(33.9)	25.49(4.60)	138.99(15.95)	1.62(1.73)
	Fifth year	10(2.2)	24.50(5.48)	137.30(14.42)	0.80(0.63)
Home residence					
	Urban	261(57.5)	25.28(4.56)	138.54(15.02)	1.45(1.51) ^a^
	Rural	193(42.5)	24.63(4.82)	137.19(16.14)	1.78(1.68)
Economic status					
	Below medium	131(28.9)	24.78 (5.13)	134.87(16.14) ^b^	1.64(0.14)
	Medium	285(62.8)	25.18 (4.48)	139.22(15.22)	1.60(1.60)
	Above medium	38 (8.4)	24.42 (4.55)	139.24(14.16)	1.16(1.26)

^a^ Significant at the 0.05 level (two-tailed). ^b^ Significant levels were corrected according to Bonferroni analysis.

**Table 2 healthcare-11-00420-t002:** The correlations among general self-efficacy and mental health.

Variables ^a^	Mental Health	Dimensions of Mental Health		
		Somatic Symptoms	Anxiety and Worry	Depression or Poor Family Relationships
General Self-Efficacy				
*r*	−0.20 ^b^	−0.14 ^b^	−0.17 ^b^	−0.13 ^b^
*p* value	<0.001	<0.001	<0.001	<0.001

^a^ The mean scores for general self-efficacy, mental health, somatic symptoms, anxiety and worry, and depression or poor family relationships were 25.00 (SD 4.68), 1.59 (SD 1.59), 0.91 (SD 0.82), 0.55 (SD 0.89), and 0.13 (SD 0.33), respectively. ^b^ Significant at the 0.01 level (two-tailed).

**Table 3 healthcare-11-00420-t003:** The correlations among mental health and psychological resilience.

Variables ^a^	Mental Health	Dimensions of Psychological Resilience							
		Social Interaction	Family Support	Friend Support	Optimistic Tolerance	Self-Worth	Self-Recognition	Self-Control	Self-Admission
Mental Health									
*r*	−0.285 ^b^	−0.119 ^b^	−0.230 ^b^	−0.137 ^b^	−0.228 ^b^	−0.303 ^b^	−0.274 ^b^	−0.171 ^b^	−0.188 ^b^
*p* value	<0.001	0.011	<0.001	0.003	<0.001	<0.001	<0.001	<0.001	<0.001
Somatic symptoms									
*r*	−0.204 ^b^	−0.105 ^b^	−0.143 ^b^	−0.101 ^b^	−0.174 ^b^	−0.187 ^b^	−0.181 ^b^	−0.121 ^b^	−0.185 ^b^
*p* value	<0.001	0.025	0.002	0.031	<0.001	<0.001	<0.001	0.010	<0.001
Anxiety and worry									
*r*	−0.246 ^b^	−0.097 ^b^	−0.206 ^b^	−0.113 ^b^	−0.194 ^b^	−0.273 ^b^	−0.246 ^b^	−0.136 ^b^	−0.149 ^b^
*p* value	<0.001	0.040	<0.001	0.016	<0.001	<0.001	<0.001	0.004	0.001
Depression or poor family relationships									
*r*	−0.206 ^b^	−0.054	−0.195 ^b^	−0.105 ^b^	−0.146 ^b^	−0.261 ^b^	−0.207 ^b^	−0.155 ^b^	−0.048
*p* value	<0.001	0.250	<0.001	0.025	0.002	<0.001	<0.001	0.001	0.309

^a^ The mean scores for resilience, social interaction, family support, friend support, optimistic tolerance, self-worth, self-recognition, self-controlling and self-admission are 137.97 (SD 15.50), 21.99 (SD 4.44), 27.27 (SD 4.21), 23.66 (SD 3.03), 14.83 (SD 2.14), 14.57 (SD 2.53), 14.58 (SD 2.23), 10.47 (SD 1.90), and 10.59 (SD 1.71) respectively. ^b^ Significant at the 0.05 level (two-tailed).

**Table 4 healthcare-11-00420-t004:** The correlations among general self-efficacy and psychological resilience.

Variables	Mental Health	Dimensions of Psychological Resilience							
		Social Interaction	Family Support	Friend Support	Optimistic Tolerance	Self-Worth	Self-Recognition	Self-Controlling	Self-Admission
General Self-Efficacy									
*r*	0.486 ^a^	0.350 ^a^	0.226 ^a^	0.286 ^a^	0.442 ^a^	0.304 ^a^	0.473 ^a^	0.380 ^a^	0.416 ^a^
*p* value	<0.001	<0.001	<0.001	<0.001	<0.001	<0.001	<0.001	<0.001	<0.001

^a^ Significant at the 0.01 level (two-tailed).

**Table 5 healthcare-11-00420-t005:** The hierarchical linear regression analysis of mental health.

Dependent Variables	Independent Variables	Standardized β
Mental health (model 1) ^a^	(1) Gender (male vs. female)	−0.075
	(2) Age (≤18 vs. 19 vs. 20 vs. 21 vs. ≥22 years)	0.018
	(3) Grade (first vs. second vs. third vs. fourth vs. fifth grade)	−0.016
	(4) Home residence (urban vs. rural)	−0.072
	(5) Economic status (medium below vs. medium vs. medium above)	−0.050
	(6) General self-efficacy	−0.201 ^d^
Psychological resilience (model 2) ^b^	(1) Gender (male vs. female)	0.202 ^d^
	(2) Age (≤18 vs. 19 vs. 20 vs. 21 vs. ≥22 years)	0.073
	(3) Grade (first vs. second vs. third vs. fourth vs. fifth grade)	−0.033
	(4) Home residence (urban vs. rural)	−0.026
	(5) Economic status (below medium vs. medium vs. above medium)	0.115 ^d^
	(6) General self-efficacy	0.516 ^d^
Mental health (model 3) ^c^	(1) Gender (male vs. female)	−0.026
	(2) Age (≤18 vs. 19 vs. 20 vs. 21 vs. ≥22 years)	0.036
	(3) Grade (first vs. second vs. third vs. fourth vs. fifth grade)	−0.024
	(4) Home residence (urban vs. rural)	−0.078
	(5) Economic status (medium below vs. medium vs. medium above)	−0.022
	(6) General self-efficacy	−0.078
	(7) Psychological resilience	−0.239 ^d^

^a^ The R^2^ and ΔR^2^ values of Model 1 were 0.054 and 0.042, respectively. ^b^ The R^2^ and ΔR^2^ values of Model 2 were 0.289 and 0.279, respectively. ^c^ The R^2^ and ΔR^2^ values of Model 3 were 0.095 and 0.081, respectively. ^d^ Significant at the 0.01 level (two-tailed).

**Table 6 healthcare-11-00420-t006:** Results from bootstrap process testing of the psychological resilience mediation model.

Model Pathways	Coefficient	95% CI	
		Lower	Upper
Mediation Effect			
General self-efficacy → Resilience → Mental health	−0.018 ^a^	−0.018 ^a^	−0.010
Direct effect			
General self-efficacy → Mental health Total effect	−0.002	−0.015	0.011
General self-efficacy → Mental health	−0.020 ^a^	−0.031	−0.011

^a^ Significant at the 0.01 level (two-tailed).

## Data Availability

Data are available from the Department of Mental Health, Brain Hospital of Hunan Province of China (contact via hnjswszx@126.com) for researchers who meet the criteria for access to confidential data.

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
