# Peer review of "The Fully Mediating Role of Psychological Resilience between Self-Efficacy and Mental Health: Evidence from the Study of College Students during the COVID-19 Pandemic"

_healthcare, 2023, doi:10.3390/healthcare11030420_

Round 1
Reviewer 1 Report
The article "Fully Mediating role of Psychological Resilience between Self-Efficacy and Mental Health: Evidence from the Study of College Student During the COVID-19 Pandemic" is interesting and treats an important topic.
However, this is an interesting subject, but I think it could have been more important if it had been presented a year ago. At this time it might have been more interesting to assess how the data taken during the pandemic changed a year later if there were repercussions for college students
Here there are other considerarions and suggestions to improve the article.
- Please explain better lines 36-37, there are no so clear.
- In several points authors wrote "a large numbers of studies" (line 39) or "many studies" (line72) or also "previous researches" and then there are only one or two studies. Please fix it. A large numbers is not two and many studies are not only one.
- Sorry, but I do not agree with the phrase in lines 44-45. First of all, what do you mean by public health worker? Nurses? Doctors? I think they were thinking about the containment of the pandemic and the healing of the sick, rather than the promotion of mental health.
- Lines 48-50 are in contrast with the previous lines (39-40).
- There are several points in which the reference is missing. For example, line 54-55, 57-58, 59-60, 60-61, 71-72, 76-78, 121-123, 268-270, 271-272. Please add it.
-How do you recruited the participants? Information about the age of participants is missing (in different countries the age for "college" could change).
-In socio-demographic information authors have to explain which information you have collected and why. Authors can not write "and so on", it is not serious in a scientific paper. In addition, were college students able to identify their family's financial status in these three categories?
- How do you calculate Cronbach's alpha? and why?
- Lines 132-135 have to be part of the results.
- What is equation 1? (line 155)
- In the note of table 1, the authors wrote the significance at two different levels. What is the significance that they show? Between which variables?
- Does GSE have a range of classifications? For example, between 20 and 26 it is s medium level?
- Line 198 could stay before table 1
- Table 5 is difficult to follow, please fix it.
Author Response
Reviewer 1
Replies to Reviewer
First of all, we sincerely thank you for your positive and constructive comments and suggestions.
Basing on the your comments and suggestions, we have revised the manuscript and now respond to you point by point as listed below. Moreover, we highlight the amendments in the revised manuscript.
With kindest regards,
Yours Sincerely,
Lu-Lu Qin and other authors
Comments and Suggestions for Authors
The article "Fully Mediating role of Psychological Resilience between Self-Efficacy and Mental Health: Evidence from the Study of College Student During the COVID-19 Pandemic" is interesting and treats an important topic.
However, this is an interesting subject, but I think it could have been more important if it had been presented a year ago. At this time it might have been more interesting to assess how the data taken during the pandemic changed a year later if there were repercussions for college students.
Answer: thank you for comments, we also care about the change of the Psychological Resilience, Self-Efficacy and Mental Health during the pandemic, which we are trying hard to study on it, especially after the implementing of the China's unlocking strategy of the COVID-19 epidemic.
Moreover, we still think it is an important findings of this our research”Fully Mediating role of Psychological Resilience between Self-Efficacy and Mental Health: Evidence from the Study of College Student During the COVID-19 Pandemic”. As everyone knows, human have been threatened by various infectious diseases. Until our research appeared, no one had studied the effect mechanism of the three variables (psychological resilience, self-efficacy and mental health) before. So we think this study will bridged the research gap and provided new horizons for research area of mental health, and a basis to improve the mental health through the mediating effect of psychological resilience.
Here there are other considerarions and suggestions to improve the article.
- Please explain better lines 36-37, there are no so clear.
Answer: thank you, we have revised it to :
College students, one of the populations who are most prone to anxiety, depression and other mental health disorders, had an unprecedented risk of growing psychological illness during the period of the COVID-19 epidemic
- In several points authors wrote "a large numbers of studies" (line 39) or "many studies" (line72) or also "previous researches" and then there are only one or two studies. Please fix it. A large numbers is not two and many studies are not only one.
Answer: thank you, we have revised it.
- Sorry, but I do not agree with the phrase in lines 44-45. First of all, what do you mean by public health worker? Nurses? Doctors? I think they were thinking about the containment of the pandemic and the healing of the sick, rather than the promotion of mental health.
Answer: thank you for your suggestive comments, we have revised it more accurate as follow :
The period of the COVID-19 requires the health department not only to heal the sick, but also to treat the mental health problem caused by the pandemic all over the world.
- Lines 48-50 are in contrast with the previous lines (39-40).
Answer: thank you, we have revised the line (39-40).
- There are several points in which the reference is missing. For example, line 54-55, 57-58, 59-60, 60-61, 71-72, 76-78, 121-123, 268-270, 271-272. Please add it.
Answer: thank you, we have added it where it is necessary.
-How do you recruited the participants? Information about the age of participants is missing (in different countries the age for "college" could change).
Answer: thank you very much, we have revised it as follow.
Besides, in China, there is no limit of the age of college students. Generally, after taking the China National College Entrance Examination, students can go to university after receiving the offer of university admission. And commonly, the age of students taking the China National College Entrance Examination are 17 or 18.
This study was a cross-sectional survey conducted in the Hunan province of China. A cross-sectional study formula was used to calculate sample size: α=0.05, n=uα/22P(1−P)/d2, where u=1.96 when α=0.05, and P was the average incidence rate of anxiety symptoms, depressive symptoms, and somatic symptoms, which is 31.53% during the period of COVID-19 [23-26]. Additionally, considering the effective rate and response rate of the questionnaire, we needed the sample size of at least 445 participants.
By using the multi-stage stratified cluster sampling, a representative sample of the college students in Hunan Province of China were selected. Firstly, one third of the 36 universities in Hunan Province are randomly selected, and the students were divided into five levels according to their grade. Secondly, two academic majors were randomly selected from each level of each school; Thirdly, one class was selected randomly from each academic major. Our trained investigators will invite students who are willing to participant in this study after introducing our research purpose and contents carefully. Data were collected through self-filling questionnaires, which were distributed and collected by researchers in the classes on site. With the approval of the Research Review Committee of the Brain Hospital of Hunan Province (NO.2020126), our this study were conducted from December 2020 to January 2021, which could be considered as COVID-19 period. Finally, a number of 480 participants from 11 academic majors' students of 12 universities were recruited.
-In socio-demographic information authors have to explain which information you have collected and why. Authors can not write "and so on", it is not serious in a scientific paper. In addition, were college students able to identify their family's financial status in these three categories?
Answer: thank you very much, we have deleted the "and so on”.
Besides, we allowed participants asking their parents to know their economic status, and we offered the per capita disposable income of residents from China National Bereau of Statistics for participants to assess their economic status.
- How do you calculate Cronbach's alpha? and why?
Answer: the Cronbach's alpha were from previous study. We have revised this in the Measures.
- Lines 132-135 have to be part of the results.
Answer: thank you very much, we have revised this.
- What is equation 1? (line 155)
Answer : equation 1: the mental health as dependent variable, and the general self-efficacy as independent variable, which describe in line 173-174.
- In the note of table 1, the authors wrote the significance at two different levels. What is the significance that they show? Between which variables?
Answer: thank you very much, we have revised this to more clearly as follow:
a Significant at the 0.05 level (two-tailed). b Significant levels were corrected according to Bonferroni analyse.
Here “a” means comparisons of the mean scores of two groups, such as GESE scores comparison between gender; “b” means the multiple comparisons of the mean scores of the three groups and above, which were corrected according to Bonferroni analyse, such as GHQ-12 Score comparisons between grades.
- Does GSE have a range of classifications? For example, between 20 and 26 it is s medium level?
Answer: as far as we know, there is no classifications of the GSE.
- Line 198 could stay before table 1
Answer: thank you, we changed it.
- Table 5 is difficult to follow, please fix it.
Answer: thank you, we have revised it more clearly.

Reviewer 2 Report
The research paper has a interesting premise and the research question is clear. The introduction could have been more elaborate and introduced the research question more thoroughly though.
There should be a designated section for the literature review and the authors should make use of relevant academic sources.
The research method is clearly described and shows good command of the main issues of the methodology, with a good take on limitations that goes beyond the usual list. However, I would have liked the authors to describe in greater detail the questions in the survey conducted.
Good analysis and synthesis, with a clear narrative reconnecting to the objectives of the study.
Author Response
Reviewer 2
Replies to Reviewer
First of all, we sincerely thank you for your positive and constructive comments and suggestions.
Basing on the your comments and suggestions, we have revised the manuscript and now respond to you point by point as listed below. Moreover, we highlight the amendments in the revised manuscript.
With kindest regards,
Yours Sincerely,
Lu-Lu Qin and other authors
Comments and Suggestions for Authors
The research paper has a interesting premise and the research question is clear. The introduction could have been more elaborate and introduced the research question more thoroughly though. There should be a designated section for the literature review and the authors should make use of relevant academic sources.
Answer: thank you for your suggestive comments, we have revised it in this manuscript.
The research method is clearly described and shows good command of the main issues of the methodology, with a good take on limitations that goes beyond the usual list. However, I would have liked the authors to describe in greater detail the questions in the survey conducted.
Good analysis and synthesis, with a clear narrative reconnecting to the objectives of the study.
Answer: thank you very much, we have improved the description of the survey in this manuscript.
Reviewer 3 Report
This is an interesting study that I appreciated reading. I have only few minor suggestions/comments:
-The authors should clearly report the aims of the study and after that the hypotheses of the study therefore line 88-98 should be corrected according to this comment
-line 98-102 are not appropriated at the end of the introduction since they represent a discussion of the possible results so I suggest to eliminate these lines or to move them in the discussion/conclusion.
-"from 24 classes of 11 majors in 12 universities" this is not clear for the readers (especially 11 majors is not clear)
- TABLE 1 grade are not clear please insert the corresponding years of education to help the reader to understand
-line 202 "The GHQ-12 score at 202 the age of 22 was lower than students of other age (P<0.01)" since there were several age groups I was wandering if the authors used some corrections as Bonferroni one for multiple comparisons
-limits I suggest to add the use of self report measures
Author Response
Reviewer 3
Replies to Reviewer
First of all, we sincerely thank you for your positive and constructive comments and suggestions.
Basing on the your comments and suggestions, we have revised the manuscript and now respond to you point by point as listed below. Moreover, we highlight the amendments in the revised manuscript.
With kindest regards,
Yours Sincerely,
Lu-Lu Qin and other authors
Comments and Suggestions for Authors
This is an interesting study that I appreciated reading. I have only few minor suggestions/comments:
-The authors should clearly report the aims of the study and after that the hypotheses of the study therefore line 88-98 should be corrected according to this comment
Answer: thank you for comments, we have revised this in the manuscript as follow:
This research hypothesized that: Firstly, self-efficacy would positively predict the mental health of undergraduate population during the period of the COVID-19 period; secondly, self-efficacy is relevant to psychological resilience; and at last, the self-efficacy of undergraduate population during the period of the COVID-19 is likely to affect their psychological health through the mediating effect of psychological resilience.
-line 98-102 are not appropriated at the end of the introduction since they represent a discussion of the possible results so I suggest to eliminate these lines or to move them in the discussion/conclusion.
Answer: thank you, we have eliminated it.
-"from 24 classes of 11 majors in 12 universities" this is not clear for the readers (especially 11 majors is not clear)
Answer: thank you, we have revised it to :11 academic majors' students from 12 universities.
- TABLE 1 grade are not clear please insert the corresponding years of education to help the reader to understand
Answer:thanks, we have changed these to : First year (1-th), Second year (2-th), Third year (3-th), Fourth Year (4-th), Fifth year (5-th).
-line 202 "The GHQ-12 score at 202 the age of 22 was lower than students of other age (P<0.01)" since there were several age groups I was wandering if the authors used some corrections as Bonferroni one for multiple comparisons
Answer: yes, we have used the analyse of Bonferroni for multiple comparisons, and we have added and revised it in the manuscript.
Table 1 :b Significant at the 0.005 level (two-tailed), Bonferroni analyse.
-limits I suggest to add the use of self report measures
Answer: thank you very much, we have added it.

Round 2
Reviewer 1 Report
I would like to thank the authors for following their suggestions and for improving the manuscript. Now, in my opinion, is ready for publication
Author Response
Thank you very much for your comments.